Emergent trees in Colophospermum mopane woodland: influence of elephant density on persistence versus attrition

O’Connor Timothy oconnortimothy55@gmail.com 1
Ferguson Angela 2
Clegg Bruce W. 3
Pallett Nita 4
Midgley Jeremy J. 2
Shimbani Julius 5
1 School of Animal, Plant and Environmental Sciences, University of the Witwatersrand , Johannesburg , Gauteng , South Africa
2 Department of Biological Sciences, University of Cape Town , Cape Town , Western Cape , South Africa
3 Department of Ecology, The Malilangwe Trust , Chiredzi , Chiredzi , Zimbabwe
4 Department of Botany, Rhodes University , Grahamstown , Eastern Cape , South Africa
5 Department of Ecology, Gonarezhou Conservation Trust , Chiredzi , Chiredzi , Zimbabwe
Cowling Richard
Electronic publication date: 2024 Feb 26
Publication date: 2024
Volume: 12
Electronic Location ID: e16961
Received 2023 Nov 16; Accepted 2024 Jan 26
Copyright: ©2024 O’Connor et al.
Copyright year: 2024
Copyright holder: O’Connor et al.
License: This is an open access article distributed under the terms of the Creative Commons Attribution License, which permits unrestricted use, distribution, reproduction and adaptation in any medium and for any purpose provided that it is properly attributed. For attribution, the original author(s), title, publication source (PeerJ) and either DOI or URL of the article must be cited.
License URL: https://creativecommons.org/licenses/by/4.0/

Keywords: Canopy volume, Gonarezhou, Hedging, Malilangwe, Ringbarking, Plant-herbivore relations, African savanna, Vegetation transformation

Funding: The Malilangwe Trust and the Gonarezhou Conservation Trust as part of their operational responsibilities Angela Ferguson was funded by The Malilangwe Trust Timothy O’Connor and Julius Shimbani were funded by the Gonarezhou Conservation Trust This work was supported by The Malilangwe Trust and the Gonarezhou Conservation Trust as part of their operational responsibilities. No external funding was involved. Specifically, the component undertaken by Angela Ferguson was funded by The Malilangwe Trust. The field component undertaken by Timothy O’Connor and Julius Shimbani were funded by the Gonarezhou Conservation Trust. The funders had no role in study design, data collection and analysis, decision to publish, or preparation of the manuscript.

==============================
Colophospermum mopane (mopane) forms mono-dominant woodlands covering extensive areas of southern Africa. Mopane provides a staple foodstuff for elephants, who hedge woodland by reducing trees to small trees or shrubs, leaving emergent trees which are too large to be pollarded. Emergent trees are important for supporting faunal biodiversity, but they can be killed by ringbarking. This study first examined the influence of elephant density on woodland transformation and the height distribution of canopy volume, and, second, whether canopy volume is maintained, and tall emergent trees too large to be broken can persist, under chronic elephant utilisation. Three regimes of 0.23, 0.59 and 2.75 elephants km−2 differed in vegetation structure and the height structure of trees. Areas under the highest elephant density supported the lowest total canopy volume owing to less canopy for plants >3 m in height, shorter trees, loss of most trees 6–10 m in height, but trees >10 m in height (>45 cm stem diameter) persisted. Under eight years of chronic utilisation by elephants, transformed mopane woodland maintained its plant density and canopy volume. Plant density was greatest for the 0–1 m height class, whereas the 3.1–6 m height class provided the bulk of canopy volume, and the 1.1–3 m height layer contained the most canopy volume. Emergent trees (>10 m in height) suffered a loss of 1.4% per annum as a result of debarking. Canopy dieback of emergent trees increased conspicuously when more than 50% of a stem was debarked, and such trees could be toppled by windthrow before being ringbarked. Thus relict emergent trees will slowly be eliminated but will not be replaced whilst smaller trees are being maintained in a pollarded state. Woodland transformation has not markedly reduced canopy volume available to elephants, but the slow attrition of emergent trees may affect supported biota, especially cavity-dependent vertebrate species, making use of these trees.

Introduction

The global biodiversity crisis demands astute custodianship of the remaining areas supporting significant natural biodiversity, of which protected areas should form the core. However, many protected areas constitute a small portion of the ecosystems they purportedly protect, such that certain natural processes may become perturbed and thereby pose a challenge for meeting the aims of protected areas. The ‘elephant problem’ is one such example for many African protected areas, in which increasing elephant densities resulting from compression of elephants into a protected area from outside areas, or accelerated rates of population growth because of improved protection, has caused dramatic changes in habitat that affects other species and ecosystem functioning (Laws, Parker & Johnstone, 1975; Lewis, 1986; Owen-Smith, 1992).

Savanna elephants are capable of transforming savanna woodlands and forest to shrubland and grassland (Laws, Parker & Johnstone, 1975; Guldemond & Van Aarde, 2008) that may cause the local extirpation of some selected species (O’Connor, Goodman & Clegg, 2007), thereby threatening the biodiversity dependent on woodland structure or on certain woody species (Cumming et al., 1997; Herremans, 1995; Nasseri, McBrayer & Schulte, 2010). The negative impact of elephants on tall trees has attracted close attention owing, in part, to the aesthetic appeal of such trees and their disproportionate value as nesting or resting sites for many animal species (Shannon et al., 2008; Chafota & Owen-Smith, 2009). When the challenges posed by elephants first emerged in the 1960s, a common response intended to restore or maintain woodland was reduction of the size of an elephant population (Wing & Buss, 1970; Laws, Parker & Johnstone, 1975). Implicit in such decisions was that elephants could be reduced to a density that would be appropriate for maintaining affected tree populations. However, the relation between elephant density and woodland structure is not well understood.

Recovery of elephant-impacted woodlands, following a change in elephant density, depends on regrowth of surviving individuals and on recruitment of new individuals. Both regrowth and recruitment are subject to ongoing impact from elephants but also additional influences including those of precipitation patterns, fire, and the impact of other herbivores (Dublin, Sinclair & McGlade, 1990; Van der Vijver, Foley & Olff, 1999; Mosugelo et al., 2002; De Beer et al., 2006). Thus, a reduction in elephant density does not necessarily ensure that recovery will occur (Laws, Parker & Johnstone, 1975). African savanna vegetation is highly diverse (White, 1983), different woody species are not accorded the same attention by elephants nor do they necessarily respond in the same way (O’Connor, Goodman & Clegg, 2007). Species-specific study of the impact on and response of individual key woody species by elephants therefore offers a sound means of improving understanding of woodland-elephant relations.

Colophospermum mopane Kirk ex Benth. (hereafter mopane) woodlands are widely distributed across the semi-arid regions of southern Africa, often forming mono-dominant woodlands (Timberlake, 1995), and falling within the previous geographic range of elephants (Coppens et al., 1978). Tall mopane woodlands may develop on well-watered, deep alluvial soils, whereas shallow, heavy-textured soils often support only a mopane shrubland (Timberlake, 1995). Mopane forms a staple of the diet of elephants where they co-occur (Guy, 1976; Villiers & De Kok, 1988; Viljoen, 1989; Lewis, 1991; Ben-Shahar, 1996; Clegg, 2010). Forms of utilisation of mopane by elephants includes stripping of leaves, breaking of small branches, pollarding (snapping) or toppling main stems in order to access foliage, uprooting, and debarking (Clegg, 2010). However, mopane is well adapted to heavy utilisation by elephants owing to its strength of coppicing (Mushove & Makoni, 1993), with stands of trees that have been reduced in height through pollarding termed ‘hedged’ mopane woodland (Styles & Skinner, 2001). By reducing the height of tall trees, thereby also reducing mean canopy height, hedging increases leaf density close to the ground that increases browse availability for elephants and for other species (Martin, 1974; Guy, 1981; Smit & Rethman, 1998; Smallie & O’Connor, 2000; Rutina, Moe & Swenson, 2005). Some tall trees may persist within hedged vegetation because their stem is too large to be pollarded or pushed over, but remaining large trees remain vulnerable to being killed through ringbarking by elephants. An unresolved question is whether increased browse availability resulting from hedging can mitigate against pollarding or debarking of remaining large trees. This is an expected outcome because pollarding and debarking of large trees is five-fold more energetically demanding for elephants than stripping of (easily accessible) leaves and twigs (Clegg & O’Connor, 2016).

An alternate prediction emerges from consideration of the foraging behaviour of elephants and the seasonal phenological cycle of mopane. Elephants, especially bulls, are preferentially grazers of soft, green grasses but consume an increasing proportion of woody material as grasses become less available during the course of the dry season (Clegg, 2010; Pretorius et al., 2011). Bulls rather than cows are responsible for most of the extreme damage to mopane trees in the form of pollarding, uprooting or debarking (Clegg, 2010). Mopane woodlands occur in environments experiencing seasonal rainfall, with a dry season of up to seven months (Timberlake, 1995). Mopane is a facultatively deciduous species, such that little mopane leaf is available to elephants at the height of the dry season (Dekker & Smit, 1996). Bulls resort to increasing their use of bark during this period despite the energetic demands of this feeding pattern (Clegg, 2010; Clegg & O’Connor, 2016) that may result in partial or complete ringbarking of trees. Partial debarking can disrupt the transport mechanisms of the tree and cause canopy dieback, whereas ringbarking will result in death or top-kill of the stem (Lewis, 1991). Although coppicing may occur at the base of a dead tall stem (Bromwich, 1972), this tall stem becomes unavailable for the many animal species which use it. Thus, remaining tall trees in a hedged woodland may continue to be impacted because the increased availability of browse through hedging is not available at the height of the dry season.

The first aim of this paper was to examine the nature and extent of impact, and patterns of utilisation, under different densities of elephant on the structure of mopane woodland. Specific questions addressed were whether an increase in elephant density resulted in (i) a decrease in the density or canopy volume of shrubs or trees, or of total canopy volume, (ii) an increase in the total amount, or proportion of the total amount, of canopy volume at a height accessible to elephants, (iii) a difference in the pattern of utilisation in terms of volume removed per plant, and degree of debarking, (iv) a trend toward elimination of tall trees, and to confirm that (v) elephants were the responsible agent for any observed differences. The second aim of this study was to determine whether, under chronic utilisation by elephants sustained over years, (i) hedged mopane woodland could maintain the amount and vertical distribution of canopy volume, (ii) tall remaining trees were spared by elephants ostensibly on account of an increased volume of available browse, or (iii) tall trees were subject to ongoing attrition from debarking, (iv) the size limit of trees that elephants can topple or pollard, and that (v) recruitment into the population should ensure population persistence.

Methods

Study area

The study was conducted in south-eastern Zimbabwe within Gonarezhou National Park (GNP) and the adjacent Malilangwe Wildlife Reserve (MWR) (Fig. 1). MWR is 394 km2 in area, and is fenced. GNP has an area of 5053 km2 and is partly fenced. The environment and vegetation of MWR has been described in Clegg & O’Connor (2012), and of GNP in Cunliffe, Muller & Mapaura (2012). Each study area experiences a similar seasonal climate of hot, wet summers and warm dry winters. Mean annual rainfall is approximately 550 mm, and daily maximum temperatures may exceed 30 °C during every month of the year. The Chiredzi River, joined by the Nyamasikana River, flows through MWR to join the Runde River, which flows west to east across northern GNP until it enters Moçambique. Woodlands in which mopane is dominant or conspicuous cover large parts of both MWR (Clegg & O’Connor, 2012) and GNP (Cunliffe, Muller & Mapaura, 2012). This study focussed on mopane woodland on alluvium, described as the Mopane Woodland on Alluvium (vegetation type 4.9) in GNP (Cunliffe, Muller & Mapaura, 2012) and as Colophospermum mopane-Courbonia glauca shrub open tall woodland in MWR (Clegg & O’Connor, 2012). This woodland type has low species richness, is dominated by mopane, and usually supports tall, emergent trees of this species.

Figure 1 Map of the study area and sampling locations.

The location of sampling plots (red dots) within Malilangwe Wildlife Reserve (n = 8) and Gonarezhou National Park (n = 7), Zimbabwe.

Data collection

The Gonarezhou Conservation Trust granted permission for this study. The study was structured as two components. The first component addressed the effects of elephant density using data collected in MWR and GNP during 2001 and 2014. The second component examined the effects of elephants at high density in GNP from 2014 to 2022.

Effects of elephant density (2001 and 2014)

The study design sought to compare this woodland type under different elephant densities, using a combination of temporal changes on fixed plots and a ‘space-for-time’ substitution. Eight plots were randomly selected using GIS along the Nyamasikana and Chiredzi Rivers in MWR, and sampled in 2001 (Clegg & O’Connor, 2012). The eight plots were relocated using a GPS in 2014, and resampled (Ferguson, 2014). Elephant density in MWR was 0.23 and 0.59 elephants km−2 in 2001 and 2014, respectively (Dunham et al., 2013), designated as ‘low’ and ‘intermediate’ density for this study. The elephant population in GNP grew exponentially from 1992 (Dunham, 2012) to attain a density of 2.75 elephants km−2 in 2013 (Dunham et al., 2013), designated as ‘high’ density for this study. Seven plots were randomly selected in GNP using GIS, and measured in 2014 (Ferguson, 2014).

Error of relocating a plot from GPS points was <5 m, further minimised by the fact that co-ordinates from all corners were initially recorded, and a detailed map of the plot was prepared at the first sampling. Any plot within MWR or GNP was >500 m from any other plot in order to ensure spatial independence.

Different sizes of woody plants in a plot were sampled using a nested transect design (Walker, 1976). A plant <3 m in height was defined as a shrub; a multi-stemmed species (e.g., Capparis spp.) was considered a shrub irrespective of its height, and otherwise a plant > 3 m in height was defined as a tree. The design employs a baseline of a fixed length of 50 m. The width of the transect was set to ensure that at least 15 individuals of the most common shrub species were sampled; belt width was then increased in order to increase the sample size of less common shrub species, and increased again to ensure more than 15 individuals of the most common tree species (mopane) were sampled. The sample size of tall, emergent mopane trees was then increased by measuring additional individual trees around a plot, and recording their positions using a GPS.

Live plants were measured if more than half the individual plant was included within the plot. The following were measured (to 0.1 m) or recorded for each live shrub or tree: (i) height, measured using a graduated rod up to 6.5 m, and estimated using the yard-stick method in increments of 0.5 m for heights >6.5 m; (ii) longest canopy diameter; (iii) canopy diameter perpendicular to the longest; (iv) canopy depth, and (v) shape of the canopy according to seven basic shapes (Fig. S1 ; Melville, Cauldwell & Bothma, 1999). For trees, the following additional measurements or estimates were taken of each stem: (i) stem circumference (cm) above the basal swelling; (ii) damage experienced by a stem, as branches lost, stem broken off, stem broken or pushed over but still attached, or stem uprooted; (iii) whether elephants or an unknown agent were responsible for damage; (iv) the volume of canopy lost, estimated using an eight-point scale (Walker, 1976; percentage classes of 0, 1–5, 6–10, 11–25, 26–50, 51–75, 76–90, 91–99, 100); and (v) occurrence of coppicing. For shrubs, only the volume of canopy lost to elephants or to other agents was estimated. Age of utilisation of woody material was defined as ‘old’ and ‘new’. New utilisation was identified by exposed wood being yellow or white, not grey, and without black splodges of algal growth, or, for bark, by seepage of gum, and had been determined to be <8 months of age. Old utilisation (>8 months) was further distinguished by coppice growth or healing of wounds. Dead stems were recorded and dead plants were defined as those with dead stems and no coppicing. Recruitment of mopane is generally considered to occur through seedling establishment (Stevens, 2021) as this species coppices from the root crown and not from roots (sensu Pausas et al., 2018); recruitment was therefore defined as stems of the smallest size class.

Elephant impacts at sustained high density: GNP, 2014 to 2022

The seven plots in GNP were resampled in 2022 (<5 m error of relocation), with elephant density having remained high at approximately 2 individuals km−2 from 2014 to 2021 (Dunham, Van der Westhuizen & Madinyenya, 2021). This provided an assessment of the impact of eight years of chronic elephant utilisation. Data collection was streamlined in order to address the main foci of (i) change in canopy volume and its height distribution, and (ii) persistence and state of large emergent trees. Measures of canopy dimensions were taken as in 2014. Repeated defoliation of a plant renders reconstruction of canopy volume unreliable. Consequently, each tree stem was scored for whether branches had been taken, the stem broken or pushed over, and, if so, whether it had continued to grow, the agent responsible, and the age of utilisation. Emergent trees were measured for the circumference and length of stem debarked, separately for old and new debarking, and their combined total was expressed as a percentage of stem circumference. The percent of canopy volume of emergent trees lost to crown dieback, easily scored because dead branches remained in the canopy, was estimated for a subset of trees (n = 19; excludes trees with zero dieback or zero debarking). Shrubs were scored for whether canopy volume had been lost to elephants or to other agents.

Plant nomenclature follows the Flora of Zimbabwe (2022).	

Data analysis

All analyses were conducted using R version 4.2.2 (R Core Team, 2022), with data manipulation and plotting done, respectively, using the ‘tidyverse’ (version 1.3.2; Wickham et al., 2019) and ‘dplr’ (version 1.0.10; Wickham et al., 2022) packages. The average of a plot is the unit of analysis unless otherwise stated.

Effects of elephant density (2001 and 2014)

The influence of elephant density (low, n = 8; intermediate, n = 8; high, n = 7) on vegetation structure was examined for the set of variables canopy volume of shrubs, trees and total; density of shrubs, live trees, live standing trees, live prostrate trees, live trees < 10 m height, live trees > 10 m height, pollarded main stems, and dead trees; mean tree height; and species density of shrubs and species richness of trees. The extent of vegetation utilisation in relation to elephant density was examined for old or new utilisation of shrubs or trees by elephants or by unknown agents. Pseudo-replication and serial correlation were unavoidable features of this landscape-level study. Accordingly, each density was tested separately against the others (i.e., low versus intermediate, low versus high, intermediate versus high). Low versus intermediate density was tested with a paired t-test, whereas a Welch’s t-test was used for the other two comparisons. The question of whether the height distribution of trees differed across the three elephant densities was examined with a chi-squared contingency test, using bin sizes of 3.1-6, 6.1-10. 10.1-14, and >14.0 m height. Canopy volume is emphasised because it represents potentially available food for elephants. Canopy volume was calculated for each height layer according to Melville, Cauldwell & Bothma (1999). Canopy volume was log10-transformed for analysis. The influence of elephant density on the distribution of canopy volume across height layers was examined using a two-way analysis of variance using the ‘car’ package (version 3.1.1; Fox & Weisberg, 2019), for which the interaction term was of primary interest.

Estimates of damage by elephants, fire, or unknown agents were derived for each height layer of a plot and the plot total. A rank for damage to a plant was first converted to the class mid-point (Walker, 1976). Loss of tree canopy biomass was weighted by the cross-sectional area of a stem, and bark damage was weighted by stem circumference. New shrub damage was weighted by the measured canopy volume of a shrub, and old shrub damage by the reconstructed canopy volume (VR) prior to change calculated as: VR = VM (100/(100 − % damage)), where VM is the measured (current) volume. (Different transect sizes were accommodated when obtaining a sum of the percentage damage for each species.) An estimate of percent damage for a plot was derived by summing these values. The influence of elephant density on the scale of damage for each damage category was examined in a pair-wise fashion, as described above.

Elephant impacts at sustained high density: GNP, 2014 to 2022

The following changes in density and canopy volume between 2014 and 2022 (n = 7) were examined. Changes in total density and total canopy volume were examined with a paired t-test. For changes in density or canopy volume per height class or height layer, an analysis of variance was undertaken, with the main effects of year (2014, 2022) and height class or layer (0–1.0, 1.1–3.0, 3.1–6.0, 6.1–10.0, >10.0); degrees of freedom were too few to permit an interaction term. These analyses were carried out in the ‘car’ (version 3.1.1; Fox & Weisberg, 2019) and ‘emmeans’ package (version 1.8.3; Lenth, 2022) for comparison of means using Tukey’s HSD. Changes in individual height classes or layers were examined with a paired t-test. The Benjamini-Hochberg correction was used to account for an increased Type 1 error rate (Benjamini & Hochberg, 1995). Canopy volume and density data were, respectively, log10- and loge-transformed for analysis. The influence of stem size on whether a stem was pollarded was examined using logistic regression, excluding stems <20 cm circumference and large stems which had been pushed over. The impact of the extent of debarking on canopy dieback of non-pollarded emergent trees was examined using logistic regression.

Results

Effects of elephant density (2001 and 2014)

Vegetation structure.

Elephant density had a marked influence on vegetation structure (Table 1). Under a sustained high elephant density in GNP, compared with low or intermediate density in MWR, tree canopy volume and total canopy volume were approximately halved; shrub canopy volume was unaffected; mean tree height was 3 m lower; the density of trees >10 m in height was more than halved, with a corresponding approximately six-fold increase in the density of pollarded stems, although the density of trees <10 m in height, and of live standing trees, were unaffected (live prostrate trees were almost absent); dead trees were two thirds less; and four and two species of shrubs and trees, respectively, had been lost. Effects of an increase from low to intermediate elephant density on Malilangwe Reserve over 14 years were apparent as a 21% loss of live standing trees, a 38% loss of trees <10 m in height, a corresponding four-fold increase in the density of pollarded main stems, a 29% reduction in shrub density, and a marginal loss of tree and shrub species.

Table 1 Structure and extent of use of Colophospermum mopane woodland under three elephant densities.

Elephant density	Low (L) (n= 8)	Intermediate (I) (n= 8)	High (H) (n= 7)	L v I	L v H	I v H	
Total shrub canopy volume (m3 ha−1)	4716 ±2402.7	2420 ±380.7	3715 ±1025.0	t =1.0263; P = 0.3389	t =0.3793; P = 0.7126	t =1.1162; P = 0.2991	
Total tree canopy volume (m3 ha−1)	23194 ±3942.0	25838 ±5765.0	11404 ±1619.0	t =0.5366; P = 0.6082	t =2.7386; P = 0.0217	t =2.3980; P = 0.0424	
Total canopy volume (m3 ha−1)	27911 ±4286.5	28258 ±5451.7	15119 ±1918.7	t =0.0563; P = 0.9567	t =2.6919; P = 0.0227	t =2.2558; P = 0.0508	
Shrub density (ha−1)	1406 ±246.0	1004 ±269.1	2100 ±811.3	t =3.3788; P = 0.0118	t =0.7699; P = 0.4667	t =1.2404; P = 0.2580	
Density of dead trees (ha−1)	32 ±6.0	24 ±3.6	9 ±3.5	t =1.3517; P = 0.2185	t =3.3899; P = 0.0057	t =2.9285; P = 0.0119	
Density live trees (ha−1)	259 ±59.3	205 ±48.0	193 ±24.1	na	na	na	
Density of standing live trees (LS) (ha−1)	258.8 ±10	205 ±48.0	193 ±24.1	t =2.3526; P = 0.0509	t =1.0194; P = 0.3333	t =0.229; P = 0.8235	
Density of live prostrate trees (ha−1)	0.5 ±0.5	0	0	na	na	na	
Density of trees <10 m height (ha−1)	220 ±67.9	136 ±46.7	178 ±23.9	t =2.7931; P = 0.0268	t =0.5734; P = 0.5805	t =0.7898; P = 0.4468	
Density of trees >10 m height (ha−1)	40 ±10.8	69 ±13.4	14 ±2.3	t =1.7408; P = 0.1320	t =2.2929; P = 0.0522	t =3.9998; P = 0.0046	
Density of pollarded main stems (ha−1)	28 ±8.6	120 ±23.9	191 ±27.6	t =5.5654; P = 0.0008	t =5.3064; P = 0.0011	t =1.8751; P = 0.0853	
Mean tree height (m)	7.9 ±0.81	9.4 ±1.47	5.1 ±0.28	t =1.8235; P = 0.1110	t =3.1826; P = 0.0113	t =2.8499; P = 0.0227	
Species richness of shrubs	11.5 ±3.2	9.3 ±3.5	7.4 ±2.1	t =2.1223; P = 0.0715	t =2.7357; P = 0.0224	t =2.2280; P = 0.0448	
Species richness of trees	3.4 ±1.3	2.4 ±0.5	1.1 ±0.4	t =1.9296; P = 0.0950	t =5.5624; P = 0.0004	t =4.0909; P = 0.0027	
Trees: Old damage by elephants (%)	5.3 ±1.0	21.3 ±3.0	50.1 ±4.3	t =6.4378; P = 0.0004	t =9.4453; P = 4.7e−05	t =5.2335; P = 0.0003	
Trees: Old damage by unknown agents (%)	9.7 ±1.7	14.3 ±2.2	5.2 ±2.3	t =1.5402; P = 0.1674	t =1.4833; P = 0.1658	t =2.7430; P = 0.0171	
Trees: New damage by elephants (%)	0.1 ±0.0	2.4 ±1.1	11.3 ±0.9	t =2.1572; P = 0.0679	t =11.921; P = 2.06e−05	t =6.3255; P = 2.65e−05	
Trees: New damage by unknown agents	0	0	0	na	na	na	
Shrubs: Old damage by elephants (%)	4.2 ±2.1	36 ±4.7	28.8 ±3.6	t =7.3440; P = 0.0002	t =5.5755; P = 0.0003	t =1.1864; P = 0.2570	
Shrubs: Old damage by unknown agents (%)	3.6 ±1.1	4.8 ±1.8	1.3 ±0.5	t =0,5591; P = 0.5935	t =1.8806; P = 0.08917	t =1.8841; P = 0.0951	
Shrubs: New damage by elephants (%)	0.1 ±0.0	6.1 ±1.7	6.6 ±0.9	t =3.6176; P = 0.0085	t =6.5194; P = 0.0006	t =0.2681; P = 0.7935	
Shrubs: New damage by unknown agents (%)	0	1.6 ±1.4	0	na	na	na	
Notes.

Differences in vegetation structure and woody species richness across Colophospermum mopane woodland areas in south-east Zimbabwe experiencing low, intermediate and high elephant density. Cell values denote mean ± SE (na denotes not applicable).

Elephant density further influenced canopy volume per height layer (Fig. 2; Table S1). The least canopy volume occurred in the 0–1.0 m layer (P < 0.05), whilst the other three height layers did not differ among themselves (P > 0.05). The smallest canopy volume under the highest elephant density shown in Table 1 was apparent only for the two height layers >3.0 m, but not for those <3.0 m in height (i.e., interaction effect). Elephant density further influenced the height structure of the tree population (Fig. 3; χ2 = 336.3; df = 6; P = 2.2e−16). Smaller trees (3.1–6.0 m in height) were well represented under all three levels of elephant density; trees from 7.1 to 11.0 m in height had been, respectively, markedly reduced or completely eliminated under an intermediate or high elephant density. There was a lower density of trees 6.1 to 14.0 m in height under a high elephant density, although a proportion of trees >14.1 m in height had persisted.

Figure 2 Impact of elephant density on canopy volume of riparian mopane woodland.

Comparison of the average canopy volume (±SE) per site (ha−1) under three different elephant densities (low, intermediate, high) and four height layers. Refer to the study area section for context on the three density regimes. Refer to Table S1 for the results of the analysis.

Figure 3 The height distribution of Colophospermum mopane trees across three elephant densities.

Frequency distribution of tree height for (A) low, (B) intermediate, and (C) high elephant density in Colophospermum mopane woodland on alluvium in south-east Zimbabwe. Refer to the Study Area section for context on the three density regimes.

Utilisation.

Consistent with an increase in elephant density was an approximately ten- and hundred-fold increase in old and new elephant damage, respectively, for trees (Table 1). Trees had lost about 25% or 60% of canopy volume to old and new elephant utilisation combined under intermediate or high elephant density, respectively, compared with 5% for the area under low elephant density. By contrast, old and new elephant damage on shrubs was considerably higher for intermediate or high elephant density than for low elephant density. Damage by unknown agents was minor, accounting for five to 14% of canopy volume, and fire damage was so slight it could be disregarded.

Areas under different elephant densities differed in terms of old or new bark damage inflicted by elephants, and for old bark damage by unknown agents, that depended on tree height (Fig. 4; Table S2). Only trees > 7 m in height were debarked to any meaningful degree. Areas under high elephant density showed a 30-fold greater level of old debarking by elephants than areas under low elephant density, but that of areas under intermediate density did not differ (P > 0.05) from either (Fig. 4A). Areas under different elephant densities differed only marginally in terms of new bark damage by elephants (Table S2), attributed to the near absence of new debarking in areas under low density (Fig. 4B). Old debarking by unknown agents was higher under an intermediate than under a low or high elephant density (Fig. 4C; Table S2).

Figure 4 Impact of elephant density on bark damage sustained by mopane trees of different height.

Differences in the percentage bark damage for trees ≤7 m, or >7 m in height, across areas supporting high (diamond), intermediate (square), or low (solid circle) elephant density for (A) old elephant debarking (OBE), (B) new elephant debarking (NBE), and (C) old unknown debarking (OBU). Note the different scale of the y-axes. Refer to Table S2 for results of the analyses.

Influence of chronic elephant utilisation in GNP: 2014–2022

Mopane contributed more than the other 26 shrub species combined to either average shrub density or average shrub canopy volume (Fig. S2), and almost all trees were mopane. The combined value of all species was therefore used for analyses. For trees and shrubs combined, no change was evident between 2014 and 2022 for either total density (t = 0.51; df = 11.81; P = 0.6208) or for total canopy volume (t = 0.8632; df = 9.8475; P = 0.4086). For the analysis of variance, there was no main effect of year (2014 to 2022) on changes in density by height class, or changes in canopy volume by height class or by height layer (Table S3). Nor were there any changes in density or canopy volume of individual height classes, or canopy volume of individual height layers (P > 0.05 for all paired t-tests) between 2014 and 2022 (Fig. 5). Although plants <1 m in height constituted most, and plants >6 m in height constituted very little of population number in either year (Fig. 5A), the bulk of canopy volume was provided by plants between 3.1 and 6.0 m in height (Fig. 5B), whereas the 1.1 to 3.0 m height layer contained the greatest amount of canopy volume (Fig. 5C). The abundance of mopane in the smallest size class (Fig. 5A) indicates this species has recruited well. Other than some dead emergent trees, dead stems and trees were conspicuously almost absent, presumably having long since been knocked over by elephants.

Figure 5 Changes in the density and canopy volume of Colophospermum mopane in Gonarezhou National Park between 2014 and 2022.

Changes of Riverine Mopane Woodland, GNP, between 2014 and 2022 (n = 7) of (A) density (ha−1) by height class, and canopy volume (m3 ha−1) by (B) height class, and (C) by height layer. Refer to Table S3 for analysis of variance tables. Superscripts indicate differences among height classes or height layers. In no case was any significant change (P < 0.05) shown for height class or height layer as tested by individual t-tests per height class or layer (loge transformed data used for density; log10 transformed data used for canopy volume).

Mopane trees had been severely utilised by 2022 (Table 2). In terms of old elephant damage, two thirds of the tree stem population had been pollarded and about 11% had escaped attention, whereas removal of branches was the main recent impact because very few stems remained to be pollarded or pushed over. The height at which a stem was pollarded was from close to ground level up to 3 m. Shrubs received less attention; only 23.4% of shrubs <1 m in height, and 80.2% of shrubs 1.1-3 m in height, had been used. The main stem of a tree was unlikely to be pollarded by elephants if it was approximately >45 cm in diameter (Fig. 6), which is the size of the emergent canopy trees >10 m in height (Fig. S3).

Table 2 Elephant damage to Colophospermum mopane trees in Gonarezhou National Park in 2022.

Elephant damage score	Percent old	Percent new	
No impact	10.9	34.9	
Branches taken	16.3	64.7	
Main stem broken	65.7	0.32	
Main stem pushed over	7.1	0	
Notes.

The percentage of the entire tree stem population (n = 312) affected by different forms of elephant damage inflicted recently (new, <8 months) or prior to that (old) in Gonarezhou National Park, recorded in 2022 (n = 7).

Figure 6 Relationship between stem size of Colophospermum mopane and pollarding.

Pollarding of large stems (>20 cm circumference) of mopane in relation to stem diameter was described by the linear logistic relationship logit (p) = 4.5596 −10.0834 x (both intercept and slope: P < 2e−16).

In summary, canopy volume was effectively maintained over an eight-year period despite chronic utilisation by elephants.

Emergent trees (n = 89) experienced attrition between 2014 and 2022, with 73% surviving intact, 15.8% having lost one or more of the main stems (>90 cm circumference), and 11.2% of trees having been lost completely. Twelve of the 15 trees with two main stems had lost the smaller stem, and two of the six trees with more than two stems had lost at least one stem. Expressing this as a stem population (n = 127) of these trees in 2022, 71.7% of the stems remained standing, and 28.3% had been lost as large stems owing to being pollarded (18.1%), falling over (8.7%), or other causes (1.6%). Percent of circumference debarked of an emergent standing tree increased on average by 25% between 2014 and 2022 (Fig. S4). The proportion of canopy volume of an emergent tree lost to dieback in relation to the extent of debarking was described by a logistic relationship (Fig. 7). Taken together, collapse of tall stems was primarily attributed to windthrow of a stem preconditioned by advanced canopy dieback once debarking exceeded 50% of stem circumference (Fig. 7). One exception was probable collapse directly from windthrow. Thus elephants indirectly caused the toppling of emergent trees and stems through canopy dieback resulting from debarking; complete ringbarking was not necessarily required.

Figure 7 The relationship between canopy dieback and the extent of debarking for Colophospermum mopane.

The relationship between the amount of canopy volume lost to canopy dieback and the percentage of stem circumference debarked is described by the logistic function y = 0.63541/(1 + exp(−0.57294∗(x − 0.034))), (adjusted R2 = 0.5970) (The two points in the bottom right were excluded).

Discussion

Elephant-mopane relations

Conversion of mopane woodland to hedged woodland by elephant utilisation has been widely reported, including in northern Botswana (Ben-Shahar, 1996), Tuli Block, Botswana (Styles & Skinner, 1997; Styles & Skinner, 2001), Luangwa Valley, Zambia (Caughley, 1976; Lewis, 1991), Kruger National Park and Limpopo Valley, South Africa (Trollope et al., 1998; Smallie & O’Connor, 2000), and, in Zimbabwe, in Sengwa Wildlife Research Area (Anderson & Walker, 1974), Hwange National Park (Boughey, 1963), and Gonarezhou National Park (Guy, 1981; Tafangenyasha, 1997; this study). In this study, conversion of riverine mopane woodland to hedged woodland depended on elephant density. In MWR, conversion of riverine mopane woodland had been initiated between 2001 and 2014 when elephant density rose from 0.23 to 0.59 elephants km−2, as evidenced by an increase in the density of pollarded stems and a loss of canopy volume (Table 1). However, the tallest height classes were maintained (Fig. 3). By contrast, riverine mopane woodlands in GNP had been converted to hedged woodland by 2014 through the loss of all trees between 6.1 m to 10 m in height, with a low density of relict tall trees too large to be pollarded (Fig. 6) remaining. Prior to the 1991/92 drought, elephant density in GNP was mostly maintained below 1.0 individual km−2 through population reduction, and was reduced to 0.8 individuals km−2 by this drought, but thereafter the population grew exponentially at 6.2% per annum to attain a density of >2 elephants km−2 in 2013 (Dunham et al., 2013). Riverine mopane woodland apparently was converted during this 23-year period. The responses observed in this study are consistent with woodland conversion becoming apparent at an elephant density of about 0.5 individuals km−2 (Cumming et al., 1997). Ground observations emphasised the role bulls play in woodland conversion, with the changes on MWR between 2001 and 2014 strongly influenced by an influx of about 70 bulls from GNP before 2014.

Conversion of tall woodland to a hedged woodland in GNP by 2014 had not decreased the amount of canopy volume under 3 m in height available to elephants (Fig. 2), nor did overall canopy volume decrease under eight years of chronic use by elephants (Figs. 5B and 5C). Furthermore, coppicing of mopane increases the availability and palatability of foliage (leaf, twig, or twig bark) (Smallie & O’Connor, 2000; Smit & Rethman, 1998; Styles & Skinner, 1997; Styles & Skinner, 2001; Hrabar, Hattas & Du Toit, 2009a) that should improve the foraging efficiency of elephants. Mopane leaf is the staple foodstuff of female elephants in south-eastern Zimbabwe (Clegg, 2010). We therefore propose that the increased availability of this foodstuff through hedging is an important influence on maintaining a high density (∼2 km−2) of elephants in a semi-arid environment.

Conversion of mopane woodland to hedged woodland has not threatened persistence of the mopane population. (A caveat is the dearth of knowledge about the effect of chronic utilisation on below-ground growth of a plant.) Elephants have transformed the growth form of an individual plant and, thereby collectively, of the vegetation structure of a woodland, but complete mortality (i.e., no coppicing) of mopane recorded in this study was relatively slight. Mortality of large trees as a result of ringbarking observed in this study is an expected result (Lewis, 1991). Reports of apparently high mortality of mopane in GNP as a consequence of toppling or pollarding by elephants (Tafangenyasha, 1997) are equivocal because trees may appear dead based on a once-off visual assessment of tree loss, but subsequent monitoring of toppled mopane trees in GNP has revealed that most affected trees coppice from the base months after impact (Bromwich, 1972). Lewis (1991) found that complete mortality of mopane trees toppled or pollarded by elephants in the Luangwa Valley, Zambia, depended on edaphic characteristics, and was precipitated by drought. Elsewhere in semi-arid, non-riparian mopane woodlands, a single drought event has caused the loss of between 4.5% and 6.9% of mopane individuals, usually smaller plants (Scholes, 1985; O’Connor, 1999), and the loss of patches of adult mopane trees on degraded habitats where water retention had been compromised (MacGregor & O’Connor, 2002). Fire does not cause conspicuous mortality of mopane because of its coppicing ability (Timberlake, 1995) and, in any event, riverine mopane woodland in Gonarezhou National Park rarely carries sufficient fuel for a burn.

Caughley (1976) proposed, based on his studies of mopane woodland and baobabs in the Luangwa Valley, Zambia, that elephant-woodland relations may follow a stable limit cycle, with a periodicity of about 200 years in the Luangwa Valley. His proposal questioned the prevailing assumption of management of that period that a stable equilibrium point exists between elephants and woodland. Putting aside theoretical and empirical challenges to his proposal (e.g., Cumming, 1982; Lewis, 1986; Lewis, 1991; Duffy et al., 1999; Baxter & Getz, 2005), we do not consider a stable-limit cycle to be an appropriate conceptual model for elephant-mopane woodland relations in south-eastern Zimbabwe. This model requires a close coupling between elephants and mopane, which apparently exists in south-eastern Zimbabwe where mopane provides the bulk of the food intake of cows (Clegg, 2010). However, this study showed that the abundance of mopane did not materially diminish even after a few decades of chronic utilisation by elephants. Furthermore, elephants show a catholic use of food species using more than 100 species in south-eastern Zimbabwe (Clegg, 2010) and comparable numbers in other systems where mopane does (Williamson, 1975; Guy, 1976) or does not occur (De Boer et al., 2000). Elephants therefore have many options for foodstuffs should mopane decline; there is no convincing evidence that elephants have a close coupling with any individual plant species.

Can relict emergent trees survive?

An hypothesis put forward for this study was that an increased availability of forage as a result of hedging would decrease use of remaining emergent trees, provided that elephant density did not continue to rise. The condition of elephant density remaining approximately stable was met (Dunham, Van der Westhuizen & Madinyenya, 2021). The elephant population remained at approximately 9,000-10,000 individuals (∼2 individuals km−2) from 2014 to 2021. The conditions were also met that hedging would not result in a decreased availability of mopane for elephants (Fig. 2), and that the amount of mopane canopy volume would be maintained under chronic elephant utilisation (Figs. 5B and 5C). However, the hypothesis was rejected on the basis of ongoing attrition of emergent trees through debarking, at a rate of 1.4% per annum lost over eight years. On MWR, elephants make greater use of debarking toward the end of the dry season when the availability of other foodstuffs has declined (Clegg, 2010). Debarking is interpreted as a foraging action of necessity rather than a preferred action because elephants require about 18 h a day to meet their foraging needs, with an adult male consuming 1–1.2% of its body weight per day (Owen-Smith, 1992), and debarking of large stems is an energetically costly process, taking about five times longer to harvest and chew a mouthful of bark from the main stem than a mouthful of leaves (Clegg & O’Connor, 2016). The time required for debarking mopane trees may be less because large sections of bark can be relatively quickly stripped off a large mopane stem once an incision has been made, compared with some tree species (e.g., Sclerocarya birrea) for which small fragments have to be tediously chiselled off (Clegg, 2010). However, bark of large mopane stems is conspicuously fibrous. Bark therefore seems less than ideal for meeting foraging needs, unless it perhaps offers an essential constituent (cf. Anderson & Walker, 1974), which has not yet been identified.

Emergent mopane trees in riverine mopane woodland, GNP, appear set to experience an ongoing slow decline in density if conditions remain similar to those recorded over the eight years of study—a high elephant density and a lack of foraging alternatives at the height of the dry season. Bulls are primarily responsible for debarking mopane, but they are preferentially grazers of green grass (Clegg, 2010, and references therein). Clegg (2010) proposed that the impact of bulls on woody vegetation would be considerably less if they had access to suitable grasslands, especially floodplain or riverine grasslands (e.g., Lewis, 1986), wetlands and reedbeds, during the dry season, which, he proposed, was the historical norm before unbridled human expansion. The far-reaching ranging patterns of elephant bulls, for which travel in the order of 50-100 km is not uncommon (summary in Dolmia et al., 2007) should enable them to access winter foraging grounds of such a nature both within and outside of GNP. Options within GNP are limited. Extensive reedbeds have occurred along portions of the Runde River within GNP that have been stripped by large floods and then regrown. For example, the flood resulting from Cyclone Eline in 2000 denuded a four kilometre stretch of reedbeds, which then recovered within about a decade, but was again stripped by Cyclone Dineo’s flood in 2017, and has not yet re-established (Appendix S1). Options outside of GNP should be substantial with the creation of the 89,000 km2 Greater Limpopo Transfrontier Conservation Area in 2002 (Ferreira, 2004) that potentially offers extensive wetlands or riverine grasslands within the Banhine and Zinave National Parks, and along the Save River, in Moçambique. To date, a current study using satellite-tracked elephants has revealed that bull elephants have begun to explore the habitat available in Moçambique, but not yet for a duration that would make a material difference to their impact within GNP (B Madinyenya, pers. comm., 2023). Their subdued use of Moçambique is attributed, in part, to the hunting pressure they encounter along the border between GNP and Moçambique (Dunham, Van der Westhuizen & Madinyenya, 2021), compounded by an increase in human settlement along the Save River and its tributaries. If these constraints change, then Moçambique may become an important foraging area for, at least, bull elephants that should diminish impact on emergent mopane trees.

Implications of structural transformation for supported biodiversity

Although an abundant mopane population has persisted in GNP in the face of chronic elephant utilisation, the potential consequences of dramatic transformation of vegetation structure for supported biota need to be considered because an aim for protected areas is to conserve all elements of indigenous biodiversity. Herremans (1995) showed an effect of elephant-transformed vegetation on avifaunal composition of comparable mopane woodland in northern Botswana, but this topic is essentially unstudied for mopane woodland. By contrast, structural change of miombo woodland in Zimbabwe by elephants had a pronounced negative effect on the richness of other plant species, birds, and some invertebrate taxa (Cumming et al., 1997). The purpose of this section is to collate selected facets of indirect evidence to propose that such biodiversity impacts are also considerable in mopane woodland, in support of further study.

Tall mopane trees possess stem cavities as a consequence of their heartwood disappearing with age (heart rot; Timberlake, 1995) which are used by a large number of vertebrate species for nesting or as a home. Many species of birds use these cavities for nesting, including hornbills, barbets, and chats (Hockey, Dean & Ryan, 2005). Tree squirrels Paraxerus cepapi are sometimes termed mopane squirrels because of their penchant for using this species as a home and place for breeding; in addition, the large seeds of mopane are an important constituent of their diet in mopane woodland (Skinner & Chimamba, 2005). On account of seed production of mopane being related to tree height (Timberlake, 1995), availability of mopane seed in hedged riverine mopane woodlands in GNP was low because only the relict emergent mopane trees, which occurred at a low density (Fig. 5), produced seed, pollarded trees did not (T O’Conner, pers. obs., 2022), although hedged mopane woodland produced seeds, albeit at a reduced amount, in Tuli Block, Botswana (Styles & Skinner, 2001). Emergent trees were also the only source of cavities, but there was a slow attrition of emergent trees primarily through debarking. The potential value of the mopane woodlands we studied for squirrels appears to have been compromised by a dramatic reduction in mopane seeds and a loss of breeding sites that is expected to impact on their numbers and on the numbers of the many predators which prey upon them. Use of tree cavities by reptiles is not as conspicuous as by birds, but some snake, skink, and agama species, as well as the rock monitor Varanus albigularis, do so (Alexander & Marais, 2007).

Caterpillars of the lepidopteran Imbrasia belina (Saturniidae; mopane worm) may exceed the biomass of elephants in semi-arid savannas during periods of irruption that affects ecosystem functioning and vegetation structure (Duffy, O’Connor & Collins, 2018; De Swardt, Wigley-Coetsee & O’Connor, 2018). However, chronic utilisation of mopane by elephants, among other factors, has been implicated in reducing their abundance or even their disappearance (Styles & Skinner, 1996; Hrabar & Du Toit, 2014), in part because of the disappearance of tall trees which are their preferred sites for laying eggs (Hrabar, Hattas & Du Toit, 2009b). Mopane worms are consumed by a suite of bird species that changes in composition as instars develop (Gaston, Chown & Styles, 1997)—reduction in the availability of mopane worms therefore has obvious ramifications for trophic patterns.

In summary, conversion of mopane woodland to hedged woodland is expected to have had far-reaching effects on the biodiversity supported by these woodlands through disruption of trophic and non-trophic linkages of mopane with faunal elements.

Conclusions

Mopane is the most resilient woody plant in the face of sustained elephant utilisation of which we are aware, on account of its coppicing ability following damage, and its high levels of recruitment into the smallest size class. Despite a decade of chronic use of riverine mopane woodland in GNP by elephants, this woody species continues to support a high plant biomass that provides a relatively stable supply of a staple elephant foodstuff. Thus there is no apparent threat to this woody species persisting, but for tall mopane trees that became relicts, only those too large to be toppled remained. We propose that relict tall trees are of disproportionate importance for maintaining faunal diversity, such that the slow erosion of their numbers over time through debarking should be contained, if possible. The conversion of mopane woodland to hedged woodland being related to increases in elephant density suggests that an adjustment of elephant density might achieve this aim. However, a traditional approach of instituting population reduction is not the only means whereby the effects of density can be adjusted; rather, ensuring access to potentially available foraging habitats, especially wetlands in which bulls can forage during the dry season, should form part of the solution.

Supplemental Information

Table S1 Analysis of variance table for the influence of elephant density and height layer on canopy volume in 2014

Table S2 Analysis of variance tables for the influence of elephant density (high, intermediate, low) and tree height (3.1–7 m; >7m) on the amount of old (OBE) or new (NBE) debarking by elephants, and old debarking by unknown agents (OBU)

Table S3 Anova table for changes in density and canopy volume between 2014 and 2022

Analysis of variance table for the influence of year and height layer on changes in density (loge transformed data) and canopy volume (log10-transformed data) of riverine mopane woodland, Gonarezhou National Park, between 2014 and 2022

Data S1 Data for the change in shrubs between 2014 and 2022 used in Figures 5A, B and C, and also in Table S3

Data not cleaned (Data S2). Refer to the figures and tables cited above.

Data S2 Change in woody canopy volume and plant density of mopane woodland in Gonarezhou National Park between 2014 and 2022

Cleaned data used in R for the derivation of Figures 5 a, b and c, as well as Table S3 and Figure S1a and S1b

Data S3 Woody height distribution shown in Figure 3, comparing Malilangwe in 2001 and 2014 with Gonarezhou in 2022

Woody height distribution based on plant height.

Data S4 Raw shrub data for Gonarezhou National Park in 2014

This data set is used in conjunction with the tree and shrub data sets for Malilangwe in 2001 and 2014, and Gonarezhou tree data in 2014, for the production of Figure 2.

Data S5 Raw data of trees in Malilangwe Reserve in 2014

This data set is used in conjunction with the tree and shrub data in accompanying files for Malilangwe in 2001 and 2014, and Gonarezhou National Park in 2014, for the production of Figures 2 and 4.

Data S6 Raw tree data for Gonarezhou national Park in 2014

This data is used in conjunction with the tree and shrub data for Malilangwe in 2001 and 2014, and the shrub data for Gonarezhou in 2014, for the production of Figures 2 and 4.

Data S7 Raw data for shrubs in Malilangwe in 2001

This dataset is used in conjunction with the shrub and tree data sets for Malilangwe in 2001 and 2014, and Gonarezhou in 2014, for the production of Figure 2

Data S8 Raw data for trees sampled in Malilangwe in 2001

This data set is used in conjunction with the datasets of trees and shrubs for Malilangwe in 2001 and 2014, and Gonarezhou in 2014 for the production of Figures 2 and 4.

Data S9 Canopy dieback of Colophospermum mopane trees in relation to debarking in Gonarezhou National Park in 2022

The data used for the production of Figure 7 based on the amount of canopy dieback of emergent mopane trees in relation to the percentage of circumference debarked in 2022.

Data S10 The likelihood of the main stem of Colophopsermum mopane being pollarded in relation to stem size, in Gonarezhou National Park, assessed in 2022

This is the data used in the production of Figure 6 and Figure S2.

Data S11 Relationship between debarking of mopane trees in 2014 and 2022

Data used in the generation of Figure S3 which shows the extent of change in the debarking of individual mopane trees in Gonarezhou National Park between 2014 and 2022

Data S12 Changes in tree density and canopy volume in Gonarezhou National Park between 2014 and 2022

This data together with the corresponding data set for changes in shrub density and canopy volume was used in the production of figure 5a, b and c, as well as Figure S1, and Table S3.

Data S13 Raw shrub data for the repeat survey of Malilangwe in 2014

All the raw shrub data of Malilangwe in 2014 to be taken together with that for Malilangwe in 2001 and Gonarezhou in 2014 for the derivation of figure 2.

Figure S1 Shapes of a woody canopy

Seven basic shapes of a woody plant canopy according to Melville, Cauldwell & Bothma (1999).

Figure S2 Contribution of 29 shrub species to density or canopy volume of the shrub layer of riparian mopane woodland, Gonarezhou National Park, Zimbabwe

Average shrub (a) density, and (b) canopy volume (n = 7) of 29 shrub species encountered. Key to species: berdis, Berchemia discolor; bosmos, Boscia mossambicensis; captom, Capparis tomentosa; cleter, Clerodendrum ternatum; colmop, Colophospermum mopane; comapi, Combretum apiculatum; commos, Combretum mossambicense; cougla, Maerua edulis (Courbonia glauca); diccin, Dichrostachys cinerea; diolou, Diospyris loureiriana; drymos, Drypetes mossambicensis; ; ehramo, Ehretia amoena; fluvir, Flueggea virosa; grelep, Grewia lepidopetala; hipcre, Hippocratea crenata; hipind, Hippocraytea indica; jasste, Jasminum stenolobum; maepar, Maerua parvifolia; maraca, Markhamia zanzibarica; pavgra, Pavetta gracillima; rhizam, Rhigozum zambeziacum; salper, Salvadora persica; senocc, Senna occidentalis; terpru, Terminalia prunioides; thiafr, Thilachium africanum; triall, Empogona kirkii (Tricalysia allenii); vitpay, Vitex payos.

Figure S3 Relationship between tree height and stem diameter for riparian mopane woodland, Gonarezhou National Park, Zimbabwe

Tree height in relation to the diameter of the main stem (n = 312; only stems with a circumference of >20 cm) is described by a logistic relationship y = 22.43158/(1 + exp(−0.47110∗(x − 0.17881))) (adjusted R2 = 0.7966).

Figure S4 Relationship between percentage debarking of a mopane stem in 2022 with that of 2014

Relationship between the proportion of stem circumference debarked in 2014 (midpoint of a percentage ranking scale) and the proportion debarked of the of the same tree’s stem in 2022 (directly measured), described by a linear relationship y = 0.53565x + 0.25674 ( F1,80 = 39.19, P = 1.7965 e −08; adjusted R2 = 0.3204).

Appendix S1 Changes in the reedbeds at Chipinda Pools

Additional Information and Declarations

Competing Interests

Author Contributions

Data Availability

The authors declare there are no competing interests. Bruce W. Clegg is employed by The Malilangwe Trust. Julius Shimbani is employed by Gonarezhou Conservation Trust. Timothy G. O’Connor is an honorary member of the School of APES, University of the Witwatersrand, and is on contract to the Gonarezhou Conservation Trust. Angela Ferguson undertook part of this work in partial fulfilment of an undergraduate degree through the University of Cape Town. Jeremy Midgley is a staff member of the University of Cape Town who served as the academic supervisor for Angela Ferguson.

Timothy O’Connor conceived and designed the experiments, performed the experiments, analyzed the data, prepared figures and/or tables, authored or reviewed drafts of the article, and approved the final draft.

Angela Ferguson conceived and designed the experiments, performed the experiments, analyzed the data, prepared figures and/or tables, authored or reviewed drafts of the article, and approved the final draft.

Bruce W. Clegg conceived and designed the experiments, performed the experiments, analyzed the data, prepared figures and/or tables, authored or reviewed drafts of the article, and approved the final draft.

Nita Pallett analyzed the data, prepared figures and/or tables, and approved the final draft.

Jeremy J. Midgley conceived and designed the experiments, analyzed the data, prepared figures and/or tables, authored or reviewed drafts of the article, and approved the final draft.

Julius Shimbani performed the experiments, prepared figures and/or tables, and approved the final draft.

The following information was supplied regarding data availability:

The raw and summary data are available in the Supplementary Files.

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
