# Peer review of "Emergent trees in Colophospermum mopane woodland: influence of elephant density on persistence versus attrition"

_PeerJ, doi:10.7717/peerj.16961_

## Round 0.1 · original submission · Minor Revisions

Both reviewers raise important points that need to be addressed in a revision. It would also be useful to speculate on the importance of below-ground architecture and biomass of mopane for the maintenance of resilience to elephant browsing. While I realise this topic was not covered in the paper, it is nonetheless likely to play an important role in our fuller understanding of savanna tree response to defoliation.

·

Basic reporting

I have a few minor concerns as marked on the review ms.

1. Recheck the formatting of the reference list, I picked up a missing full-stop, line 675.

2. Figure 2: The colour-coding does not match with figure title/label description

Experimental design

no comment

Validity of the findings

Line 316/317: "In summary, canopy volume was effectively maintained over an eight-year period despite chronic utilisation by elephants."

I am having trouble with understanding this interpretation of the results or the phrasing of this statement.

The eight-year period referred to here represents the 2014-2022 period sampled and thus the categories referred to as from Intermediate to High Utilization (as set out in the methods section) ?

Table 1 shows a statisitically significant regression in total canopy volume between elephant density categories : Low vs High and Intermediate vs High density. Only for the shift from Low to Intermediate is there no significant change in total canopy volume.

Difference in total shrub (all multistemmed individuals and trees < 3m) canopy volume remained insignificant (Table 1) - and thus assuming that for elephant optimal feeding size are what is defined here as shrubs, then this statement is true.

I suggest it needs to be qualified as referring to shrubs or as available browse for elephants (as being one of the hypotheses discussed in the Discussion section), otherwise it contradicts line 253 which states "tree canopy volume and total canopy volume were approximately halved".

I wonder how much and if significant biomass loss (above and belowground) is implicated in this halving of canopy volume between low and high elephant densities.

Additional comments

The study is considered as being well conceived and executed. From the data provided to the final results, it succeeds in giving some answers to the questions it sets out to investigate in a professional manner that is based on sound analysis of a relatively robust dataset . The study will likely represent an useful example on how impacts such as sustained elephant density and likely also charcoal production on alluvial tall Mopane woodland vegetation can be quantified and monitored henceforth.

·

Basic reporting

This manuscript explores the physiognomic changes is riverine mopane savanna driven by elephants.
To do this, it uses two repeat-measurement datasets in different areas. The manuscript is well-written and relevant.
My (limited) understanding of C. mopane is that it is a root-suckering sobole (sensu Pausas et al., 2018: doi: 10.1111/nph.14982). How were recruits via root-suckers differentiated from seedlings? (i.e. Ln 189, Ln 305). Is recruitment considered to be via root-suckers or seedlings, but not differentiated in this study? If so, please make that clear.
One thing that I felt was lacking fairly often for the reader were sample sizes. Please provide wherever possible (e.g. Ln 201).
There is an issue of elephant density, which is calculated by the total population divided by the reserve area. However, this assumes even utilisation across the area. This is very unlikely here given the study system is near water. Please highlight the qualitative nature of the elephant utilisation/density categories to the reader.

Experimental design

I find the methods descriptions quite confusing, and in need of clarification and elaboration:
How certain are you that the exact same plots were used? Were they physically marked? How were GPS points used for the transects? i.e. it sounds like a single GPS point was used to locate a “plot” but it’s actually a transect — how was it ensured that the same area was covered? A change in direction from the starting to ending point could change the trees sampled quite substantially, and thus this can greatly influence the fixed-point temporal comparison.
The terminology of plot vs belt transect needs to be standardised. Please use one or the other.
Ln. 167. A belt transect of “sufficient width”. The area is available in raw data, so widths were kept track of, but in the text, this is not clear. Make it clear to the reader that keeping track of area was important and widths were recorded, not arbitrarily decided upon to get sufficient numbers. Also, it’s not clear if area was kept constant for plots between time periods. Please put some effort into clarifying plot sampling to the reader.
How were emergent trees outside of the belt transect tracked? Not clear.
Ln. 236. I don’t understand how this aov could have been setup. What is the response variable? What is the predictor variable? Counts in each class? I suspect a simple Chi-squared test would be more appropriate here.
Ln. 318. Were these emergent trees tagged? What happens if they were lost because the plot had been shifted?

Validity of the findings

As long as plot positioning was constant, then I am sure the findings are valid. However, shifting plots could possibly result in substantial changes given the small number of plots per time period/area (i.e. added spatial variability on top of temporal variability). The authors need to clarify this to the reader.

Additional comments

Ln 152. Is this not a ‘time-for-space’ substitution?
Ln 178. Please provide a brief description of the basic shapes to the reader. A photo-based image in the appendix would be ideal.
Ln 182. Similar here for the eight point scale. Please provide a description for readers.
Ln 183. Coppicing and resprouting are used fairly synonymously in the text, but I think there are two types of resprouting going on here — coppicing and root-suckering. I think this should be clarified in the text. (Coppicing — ramets come up in a centralised area; root-suckering — ramets come up at varying distances from the main stem).
Ln 206. Please provide version of R. And elsewhere, the version numbers of the libraries used.
Ln 240. Not RStudio. Analyses were conducted in the R environment. You can cite RStudio as a GUI interface, but analyses are conducted in R.
Ln 406. This information on elephant population sizes needs to be earlier in the paper.
Ln 421. “Off” to “of”
Ln 496. Could very well be root-suckering regeneration, especially as the study system appears to be in alluvial floodplain soil.
Figure 2 – “solid/dotted/hatched” are not shown in the figure, rather lightgrey, darkgrey and black. Please include a legend in the figure.
Figure 4. I don’t understand what the solid lines going off the left-hand side of each panel represents. Remove?
Figure 5. Sure dissimilar superscripts indicate significant differences. How were these obtained? A post-hoc Tukey test? Or lots of t-tests??? Were the tests conducted lumping 2014 and 2022 data? It’s very much not clear. Also, are plots going into these boxplots? Or individual trees? This figure legend and analyses needs far more explanation as to what it represents.

---

## Round 0.2 · accepted · Accept

Both reviewers are satisfied that their comments have been addressed in the revised version. Reviewer 2 does, however, provide a few minor comments that can be addressed without submitting another draft. Your paper is now ready for publication.

·

Basic reporting

Please mind the resolution of images in final publication as they are not great in this submission, otherwise no comment.

Experimental design

No comment

Validity of the findings

No comment, my initial confusion has been clarified with the new version.

Additional comments

The authors clarified the initial misinterpretation and confusion on my side in this revised version.

·

Basic reporting

The authors have greatly improved the flow of the manuscript, and it is far easier to follow the detailed work that was conducted.

I have three minor points for consideration

In the abstract...
Ln 28: Areas with highest elephant density had lowest canopy volume.
Ln 39: not markedly reduced canopy volume available to elephants...?
These appear to be conflicting statements. Maybe an addition to ln 28 - "; but this was only marginal"?

Ln 163: Is it possible to include a sentence about what is considered a low or high density of elephants? (As PeerJ has a broad readership - I get that 2.7 elephants per sqkm is a lot, but that might not be clear to a general reader).

Ln 198/199 - stems of the smallest size class occurring singly and away from pollarded main stems, yes? Stems of the smallest size class is still too vague as that could include resprouts from the root crown.

Experimental design

A good experimental design which shows robust results.

Validity of the findings

Findings are valid, with suitable speculation of the results in the discussion.